# Unraveling the Guardians of Growth: A Comprehensive Analysis of the *Aux*/*IAA* and *ARF* Gene Families in *Populus simonii*

**DOI:** 10.3390/plants12203566

**Published:** 2023-10-13

**Authors:** Kewei Cai, Qiushuang Zhao, Jinwang Zhang, Hongtao Yuan, Hanxi Li, Lu Han, Xuebo Li, Kailong Li, Tingbo Jiang, Xiyang Zhao

**Affiliations:** 1State Key Laboratory of Tree Genetics and Breeding, Northeast Forestry University, Harbin 150040, China; ckwnefu@163.com (K.C.); zhaoqs1002@163.com (Q.Z.); 17644249340@163.com (H.L.); likailong@126.com (K.L.); tbjiang@nefu.edu.cn (T.J.); 2Tongliao Forestry and Grassland Science Research Institute, Tongliao 028000, China; nmtlzjw@163.com (J.Z.); 19997161499@163.com (H.Y.); 3Jilin Provincial Key Laboratory of Tree and Grass Genetics and Breeding, College of Forestry and Grassland Science, Jilin Agricultural University, Changchun 130118, China; hl13596256430@163.com; 4Changling County Front Seven State-Owned Forest Protection Center, Changling 131500, China

**Keywords:** *Populus simonii*, *Aux/IAA* family, *ARF* family, gene expression pattern, protein–protein interaction

## Abstract

The auxin/indole-3-acetic acid (*Aux/IAA*) and auxin response factor (*ARF*) genes are two crucial gene families in the plant auxin signaling pathway. Nonetheless, there is limited knowledge regarding the *Aux/IAA* and *ARF* gene families in *Populus simonii*. In this study, we first identified 33 putative *PsIAAs* and 35 *PsARFs* in the *Populus simonii* genome. Analysis of chromosomal location showed that the *PsIAAs* and *PsARFs* were distributed unevenly across 17 chromosomes, with the greatest abundance observed on chromosomes 2. Furthermore, based on the homology of *PsIAAs* and *PsARFs*, two phylogenetic trees were constructed, classifying 33 *PsIAAs* and 35 *PsARFs* into three subgroups each. Five pairs of *PsIAA* genes were identified as the outcome of tandem duplication, but no tandem repeat gene pairs were found in the *PsARF* family. The expression profiling of *PsIAAs* and *PsARFs* revealed that several genes exhibited upregulation in different tissues and under various stress conditions, indicating their potential key roles in plant development and stress responses. The variance in expression patterns of specific *PsIAAs* and *PsARFs* was corroborated through RT-qPCR analysis. Most importantly, we instituted that the *PsIAA7* gene, functioning as a central hub, exhibits interactions with numerous Aux/IAA and ARF proteins. Furthermore, subcellular localization findings indicate that *PsIAA7* functions as a protein localized within the nucleus. To conclude, the in-depth analysis provided in this study will contribute significantly to advancing our knowledge of the roles played by *PsIAA* and *PsARF* families in both the development of *P. simonii* tissue and its responses to stress. The insights gained will serve as a valuable asset for further inquiries into the biological functions of these gene families.

## 1. Introduction

The plant hormone auxin, also known as plant growth hormone or IAA (indole-3-acetic acid), is one of the essential plant hormones. It is a naturally occurring compound synthesized in plant cells, predominantly present in the apical buds, root tips, young leaves, and fruits of plants [1]. Moreover, it influences various parts of the plant through intercellular signal transmission. Auxin/indole-3-acetic acid (*Aux/IAA*) and auxin response factor (*ARF*) families serve as core components in the signaling pathway of auxin, garnering significant attention in this field. They are involved in regulating the plant’s response and perception of auxin and play essential roles in the development of various plant organs as well as the plant’s adaptation to changes in the external environment [2].

The *AUX/IAA* and *ARF* families are known for their distinctive structural features, which are critical for their functions in auxin signaling and transcriptional regulation. Typical Aux/IAAs are characterized by four specific regions (domains I–IV) that contribute to their functional properties as short-lived nuclear proteins. Domain I is particularly significant as it contains the repeat motif represented by “LxLxL”, enabling interactions with other essential proteins involved in auxin signaling pathways. Through interactions within this domain, *AUX/IAA* gene family members act as transcriptional repressors, modulating the expression of downstream target genes [3]. Domain II contains a degron sequence, allowing rapid degradation of the gene product in response to changes in auxin levels, thereby tightly controlling the duration and intensity of auxin signaling [4]. Domain III, termed the “EAR motif” functions as a transcriptional repression domain, contributing to the regulation of auxin-responsive genes [5]. Lastly, Domain IV comprises a PB1 dimerization motif that facilitates protein–protein interactions, enabling the formation of functional protein complexes [6]. The *ARF* family has three domains that control its functional properties. One of the key domains is the DNA-binding domain (DBD) located at the N-terminus. This domain allows ARF proteins to bind to specific auxin-responsive elements in the promoters of target genes, thus regulating their expression in response to auxin signaling. Another essential region is the middle region, also called the middle domain (MD). This domain participates in transcriptional activation or repression, depending on the specific ARF member and its interaction partners. ARF proteins also contain a C-terminal dimerization domain (CTD) responsible for the formation of homodimers or heterodimers with other transcription factors [7]. This dimerization capability influences their transcriptional activity and interaction with other proteins in auxin signaling pathways. The combined effects of these characteristic regions in the *AUX/IAA* and *ARF* family allow for precise control over auxin signaling, orchestrating essential processes throughout plant development.

In plants, auxin exerts its effects primarily by regulating its synthesis, transport, and signal transduction, with the Aux/IAA–ARF module playing a crucial role as a key component of auxin signal transduction [8]. Members of the *ARF* family form ARF–IAA complexes by binding to *IAA* inside the cells. When ARF binds to IAA, the transcription factor activity of *ARF* is repressed, leading to the inhibition of specific gene expression [1]. This mechanism enables plants to fine-tune gene expression levels in response to both environmental conditions and internal signals, allowing them to adapt to varying growth conditions. However, when plants encounter external stress or reach specific developmental stages, members of the *IAA* family can be degraded, causing the dissociation of the ARF-IAA complex. Consequently, the transcription factor activity of *ARF* is released, initiating the expression of specific genes, which triggers a series of growth and developmental responses [9,10]. In summary, the interplay between *ARF* family and *IAA* family is a complex and tightly regulated system. They jointly participate in the regulation of plant growth and development, enabling plants to adapt to variable environmental conditions and ensure their survival and reproduction.

Previous studies have confirmed that members of the *Aux/IAA* and *ARF* families play significant roles in various developmental processes and resistance responses in multiple plant species, such as *Arabidopsis*, *Gossypium hirsutum*, and *Oryza sativa*. For example, the mutant axr3 (iaa17) of the *Aux/IAA* family member *AXR3* in *Arabidopsis thaliana* exhibits noticeable phenotypic changes, including a significant reduction in hypocotyl length, and upward curling and darkening of leaves [11]. *SlIAA9* is involved in tomato fruit development and leaf morphogenesis, its downregulation leads to the transformation of compound leaves into simple leaves, and transgenic lines with suppressed expression of *SlIAA9* exhibit asymmetric sepals [12]. In *Eucalyptus grandis*, *EgrIAA4* acts as an auxin response inhibitor, and its overexpression in *Arabidopsis* significantly impairs plant growth and reproduction, inhibiting primary root elongation and lateral root initiation [13]. Furthermore, *OsIAA20* in rice plays a crucial role in response to abiotic stress, and under drought and salt stress conditions, the *OsIAA20* RNAi rice line exhibits a significant decrease in proline and chlorophyll content, along with a significant increase in malondialdehyde content [14]. Furthermore, research has revealed that the *AtARF1* and *AtARF2* transcriptional repressor factors in *Arabidopsis thaliana* play pivotal roles in the regulation of leaf senescence, floral organ abscission, and cell proliferation [15,16]. The candidate genes *GbARF10b* and *GbARF10a* were validated as transcriptional activators in Ginkgo male floral organ development and their involvement in the IAA signaling pathway, as well as their response to exogenous SA [17]. In peanut, the *ARF* gene *Arahy.7DXUOK* was identified as a potential target of miR160, and its transcript degradation through miR160 overexpression in transgenic plants resulted in increased salt tolerance [18]. Furthermore, in a study by Wang et al. [19], yeast two-hybrid experiments were employed to demonstrate that *AtIAA8* interacts with *ARF6* and *ARF8*, consequently influencing the expression of downstream auxin-inducible genes and thereby regulating the development of floral organs in *Arabidopsis*. The *Aux/IAA* and *ARF* families have been extensively characterized and described in important species such as rice [20], *Populus trichocarpa* [21], soybean [22], sweet orange [23], and moso bamboo [6], because they are involved in various physiological and developmental processes and hold the potential to enhance species resistance. *Populus simonii* is a key native tree species in northern China, it possesses strong stress resistance, wide adaptability, and excellent hybrid compatibility. It is currently the primary tree species used for afforestation and ecological construction in the “Three-North” regions of China and serves as crucial parent material for cultivating superior stress-resistant clones [24]. Due to the diverse functions exhibited by the *Aux/IAA* and *ARF* families across various plant species, it becomes imperative to explore their implications within *P. simonii*. In this study, we conducted a comprehensive investigation of the *Aux/IAA* and *ARF* families in *P. simonii*, including the number of family members, chromosomal location information, gene structure, evolutionary relationship, tissue-specific expression patterns, protein–protein interaction (PPI) and qRT-PCR analysis, as well as subcellular localization analysis. Through an in-depth investigation of the *AUX/IAA* and *ARF* families, we aim to unravel the precise mechanisms through which they are involved in auxin regulation. This will provide new perspectives and strategies for studying growth and development in *P. simonii*.

## 2. Results

### 2.1. Identification of PsIAA and PsARF Genes

By performing an ortholog blasting search against *P. simonii* using the known AUX/IAA and ARF protein sequences of *A. thaliana*, a comprehensive set of 33 and 35 protein sequences were identified and confirmed as PsIAA and PsARF family members in *P. simonii*. Subsequently, the 33 *PsIAA* and 35 *PsARF* genes were named in the order of homologous chromosomes as *PsIAA1* to *PsIAA33*, and *PtrARF1* to *PtrARF35*, respectively (Appendix A). Next, to enhance our comprehension of the attributes of *PsIAA* and *PsARF* genes, we examined fundamental details such as protein length, projected isoelectric point (pI), molecular weight (MW), and other essential information, as outlined in Appendix A. The findings reveal that PsIAA proteins typically consist of 153–367 amino acids, with a pI ranging from 4.86 to 9.25, a MW ranging from 17.091 to 39.858 kDa, and instability index from 26.46 to 57.04. However, in PsARF proteins, the length of all PsARF protein sequences ranged from 592 to 1226 amino acids, with the average length of PsARF proteins being notably higher than that of PsIAA proteins. Furthermore, the PsARF proteins exhibited molecular weight (MW) values spanning from 65.314 to 136.919 kDa, while their isoelectric points (pI) ranged from 5.28 to 8.94.

### 2.2. Phylogenetic Analysis of PsIAA and PsARF Genes

To explore the evolutionary connections, we conducted distinct multiple sequence alignments for both *PsIAA* and *PsARF* families. Subsequently, phylogenetic trees were crafted individually for each of these families (Figure 1). The 33 *PsIAA* genes in the constructed phylogenetic tree were categorized into three subgroups, namely IAA-A, IAA-B, and IAA-C, based on the homology of their proteins (Figure 1a). The highest count was observed in IAA-C, encompassing a total of 14 genes, with IAA-A coming next in terms of gene count. The majority of PsIAA proteins exhibited four consistently conserved domains (I–IV). The four proteins (PsIAA20, PsIAA32, PsIAA8, and PsIAA30) lack domain I, and are all distributed within one subfamily. Moreover, in the constructed phylogenetic tree, the 35 *PsARF* genes were also divided into three subgroups (including ARF-A, ARF-B and ARF-C) (Figure 1b). ARF-C, which includes 15 *PsARF* genes, represents the largest subgroup in terms of gene count. Within every subfamily, the PsARF proteins displayed the conserved DBD and MR domains; nevertheless, 21 PsARF proteins were devoid of the CTD domain. This absence of the CTD domain may render them incapable of interacting with Aux/IAA and they may be insensitive to auxin.

### 2.3. Gene Structure, Motif Composition of PsIAA and PsARF Genes

Exon–intron analysis was conducted on *PsIAA* and *PsARF* genes to comprehend their structural diversity. This finding unveiled the presence of introns within all genes, with the 33 *PsIAAs* and 35 *PsARFs* exhibiting diverse numbers of exons (Figure 2). Specifically, the *PsIAA* genes had exons ranging from 2 to 6, while the *PsARF* genes had exons ranging from 2 to 15. Additionally, a significant portion of genes within each category displayed comparable gene architectures, whether in relation to exon lengths or intron quantities. Notably, eight *PsARFs* exclusively contain two to four exons, and all of these genes are found only in ARF-C. While most genes within the same subfamily exhibited similar intron/exon numbers, significant differences were also observed in certain subfamilies. As an example, considerable structural disparities, along with variations in the number of introns and exons, were observed in *PsIAA2/-7*, *PsIAA19/-32*, and *PsARF1/-15*. Moreover, *PsIAA12*, *PsARF1*, and *PsARF1* displayed a distinct evolutionary trajectory, characterized by the presence of exons and introns but devoid of the untranslated region (UTR).

For the exploration of functional domains within PsIAA and PsARF proteins, amino acid motifs were scrutinized by employing the MEME online tool (Figure 2). The clustering of the majority of *PsIAA* and *PsARF* genes into a single clade reveals a shared motif composition at the same position, pointing to their potential similarity in biological functions. Nevertheless, notable discrepancies in motif composition were discernible across diverse categories. For example, the ARF-A and B clusters exhibited the greatest quantity of motifs and the most intricate arrangement of motifs. Furthermore, certain motifs exhibited specificity to distinct groups. As an illustration, motif 10 was exclusively detected within the *PsIAA* family, whereas in the *PsARF* family, motif 8 was specific to the ARF-A and B groups. Among the ten scrutinized motifs, motifs 2, and 5 were shared by both the *Aux/IAA* and *ARF* families.

### 2.4. Cis-Elements in the Promoters of PsIAA and PsARF Genes

For the exploration of the biological roles of *PsIAAs* and *PsARFs*, the cis-elements located within the 2 kb upstream sequences of each respective gene were analyzed using the PlantCARE Database. A total of 13 distinct types of cis-elements were identified within the upstream promoter regions of both *PsIAAs* and *PsARFs* (Figure 3). Among these elements, many were associated with abiotic stresses, encompassing defense and stress responsiveness, drought-inducibility, and low-temperature responsiveness. The elements related to plant growth and development included meristem expression, circadian control, and light responsiveness. The elements associated with hormone responses included auxin responsiveness, MeJA responsiveness, gibberellin responsiveness, abscisic acid responsiveness, salicylic acid responsiveness, and zeatin metabolism regulation. Moreover, several elements exhibited responses to specific stresses. For instance, Box 4, MRE, AE-box, I-box, and G-Box were associated with light responsiveness, while AuxRE, TGA-element, and AuxRR-core were linked to auxin responsiveness. Additionally, P-box, GARE-motif, and TATC-box were found to be involved in gibberellin responsiveness. This diversity of cis-elements suggests that *PsIAA* and *PsARF* genes play roles in numerous biological processes, serving as connectors between multiple hormone responses and other vital biological pathways.

### 2.5. Chromosomal Location and Synteny Relationship of PsIAA and PsARF Genes

Using the genomic data of *P. simonii*, we conducted an analysis of the chromosomal distribution patterns of *PsIAA* and *PsARF* genes. The results of the chromosome localization analysis revealed an uneven distribution of 33 *PsIAA* and 35 *PsARF* genes across 17 out of the total 19 chromosomes of the *P. simonii* genome (Figure 4a). Notably, no *PsIAA* and *PsARF* genes were found to be anchored on chromosomes 7, and 19. Among the chromosomes, chromosome 2 exhibited the highest abundance with ten *PsIAA* and *PsARF* genes, while surprisingly, the largest chromosome, Chr01, only contained seven *PsIAA* and *PsARF* genes. Interestingly, chromosomes 4, 9, 11, 12, 15, 16, and 17 exclusively carried *PsARF* genes, while chromosome 13 solely contained *PsIAA* genes. Within the *PsARF* family, no tandem duplicated gene pairs were detected, as these genes are separated by considerable megabase intervals. However, in the *PsIAA* family, we identified five pairs of tandem duplicated genes, distributed on chromosomes 1, 5, 8, and 10, respectively.

The Advanced Circos program in TBtools was employed for conducting synteny analysis, aiming to explore the evolutionary relationship among all *PsIAA* and *PsARF* genes (Figure 4b). The analysis results reveal 36 homologous gene pairs among the 33 *PsIAA* genes and 30 homologous gene pairs among the 35 *PsARF* genes, indicating a significant and important interrelationship between them. Furthermore, syntenic genes were most prevalent on chromosome 2 (16 gene pairs), followed by chromosome 1 and chromosome 5 (11 gene pairs). It is worth noting that some *PsIAA* and *PsARF* genes exhibit correspondence with multiple genes located on other chromosomes.

### 2.6. Synteny Relationships of IAA and ARF Genes in P. simonii and Different Species

For the analysis of *PsIAA* and *PsARF* gene families’ evolution, we generated a comparative syntenic diagram involving *P. simonii* and two representative species (including *Populus trichocarpa*, and *A. thaliana*) (Figure 5). In the *PsIAA* family, out of 33 *PsIAAs*, 64 and 108 homologous gene pairs were identified with *Arabidopsis* and *P. trichocarpa*, respectively. Among these, the majority of homologous gene pairs with *Arabidopsis* and *P. trichocarpa* were distributed on chromosome 5 of *P. simonii* (with 12 and 22 pairs, respectively). However, in the *PsARF* family, the number of homologous gene pairs with *Arabidopsis* and *P. trichocarpa* was smaller compared to the *PsIAA* family. Out of 35 *PsARF* genes, there were 24 and 87 homologous gene pairs with *Arabidopsis* and *P. trichocarpa*, respectively. The largest number of gene pairs was located on chromosome 2 of *P. simonii*. As expected, the homology of *IAA* and *ARF* genes between *P. simonii* and *P. trichocarpa* was higher compared to *Arabidopsis*, which may be related to species evolution and genetic relationship. Furthermore, our examination of both intra- and inter-species synteny, focusing on *PsIAA* and *PsARF* genes, unveiled that chromosome 2 and 5 exhibited the highest abundance and diversity of syntenic gene pairs.

### 2.7. Regulatory Network and Expression Patterns of PsIAA and PsARF Genes

An interaction network comprising all *PsIAA* and *PsARF* genes was established through the use of the STRING website tools, based on their homology to proteins in *P. trichocarpa*, which aimed to delve deeper into the potential connection of these genes (Figure 6a). The results revealed that among the 68 *PsIAA* and *PsARF* genes, 58 of them were involved in constructing a complex gene regulatory network. The majority of these genes functioned as hub genes within the regulatory network, for instance, *PsIAA7* acted as a hub gene interacting with 35 other genes, while *PsARF18* functioned as a hub gene interacting with 30 other genes. The identification of regulatory networks provides valuable information for better understanding the roles of *IAA* and *ARF* genes in development, and stress responses.

To investigate the roles of *PsIAA* and *PsARF* genes in *P. simonii*, we utilized RNA-seq data to examine the expression patterns of *PsIAA* and *PsARF* genes across six distinct tissues, namely, terminal buds (NT), axillary buds (NB), leaves (NL), stems (NS), phloem (NP), and roots (NR). Distinct tissue-specific expression patterns were observed for *PsIAA* and *PsARF* genes (Figure 6b). For instance, *PsIAA19* exhibited high expression exclusively in the phloem, while its expression levels were comparatively low in other tissues. *PsIAA1*, *PsIAA3*, and *PsIAA14* showed only low expression levels in leaves, while higher expression patterns were observed in other tissues. Furthermore, *PsARF17*, *PsARF21*, *PsARF33*, *PsARF12*, and *PsARF3* exhibited relatively low expression levels in both leaves and roots. It is noteworthy that we observed some *PsIAA* and *PsARF* genes with consistently high expression levels across all tissues, such as *PsIAA7*, *PsIAA29*, *PsARF28*, and *PsARF31*.

Since we have identified stress-responsive cis-elements in the *PsIAA* and *PsARF* gene promoters, we further investigated the expression patterns of these genes using RNA-seq data from leaves of *P. simonii* under heat stress, cold stress, and salt stress conditions (Figure 6c). Most of the *PsIAA* and *PsARF* genes exhibited lower gene expression levels under various stress conditions. Nevertheless, a minor subset of genes exhibited noteworthy alterations in expression levels under varying stress conditions, manifesting elevated gene expression levels. For example, *PsIAA20* showed higher gene expression patterns under salt stress and heat stress conditions, while *PsIAA26* exhibited elevated gene expression levels under cold stress. Moreover, *PsIAA7*, *PsIAA18*, *PsARF13*, *PsARF3*, and *PsARF30* responded to all stress conditions, demonstrating consistently high gene expression levels. Interestingly, *PsIAA7* serves as both a hub gene in the regulatory network and a gene with high expression levels across various tissues and stress responses. On the other hand, another hub gene, *PsARF18* exhibited lower expression levels across all tissues and stress responses compared to *PsIAA7*, suggesting that *PsIAA7* may play a more crucial role in plant growth and development, and stress responses.

### 2.8. Quantitative Reverse Transcription Polymerase Chain Reaction (qRT-PCR) Analysis

In order to enhance our comprehension of the expression profiles of *PsIAA* and *PsARF* genes across diverse tissues, we chose six *PsIAAs* and six *PsARFs* at random and conducted qRT-PCR analysis using tailored primers. The expression patterns of these *PsIAA* and *PsARF* genes across various tissues closely align with the previously established expression profiles of these genes, indicating the validity of utilizing RNA-seq data to assess the expression levels of these genes (Figure 7). Furthermore, the correlation coefficient between the RNA-seq data and qPCR data was 0.8067, providing additional evidence for the reliability of transcriptomic data (Appendix A).

### 2.9. Subcellular Localization of PsIAA7

From the preceding heatmap analysis of distinct tissues and stress treatments, it was evident that a hub gene member (*PsIAA7*) consistently displayed high expression levels. Hence, *PsIAA7* should be regarded as a noteworthy candidate gene with distinct functions pertaining to auxin signaling transduction. To further authenticate the central role of *PsIAA7*, we opted to conduct subcellular localization experiments focusing on this hub gene. The effective pCAMBIA1300-35S:*PsIAA7*-GFP fusion expression vector was successfully generated and introduced into tobacco leaves through Agrobacterium GV3101-mediated transformation. In Figure 8, the transformed tobacco leaf cells expressed *PsIAA*-GFP fusion proteins, and the fluorescence signal was exclusively observed in the cell nucleus, indicating their function as nuclear-localized transcription factors in regulating relevant biological processes.

## 3. Discussion

The *IAA* and *ARF* gene families constitute integral components of the plant hormone auxin signaling pathway, holding pivotal functions in plant growth, morphogenesis, hormonal responses, and adaptation to stress. *PsIAA* and *PsARF* genes in *P. simonii* have not been subjected to genome wide analysis, and their regulatory functions remain unclear, despite the identification and characterization of some *IAA* and *ARF* family members in other plant species. This study encompassed a genome-wide investigation and expression analysis of the *IAA* and *ARF* families, providing genetic insights that advance our comprehension of growth, development, and stress responses in *P. simonii*.

The size and structural characteristics of gene families are influenced not only by a species’ genome size but also by the intricate evolutionary processes within plant species [25]. Within this investigation, a cumulative count of 33 *Aux/IAA* genes and 35 *ARF* genes were successfully identified within the *P. simonii* genome. When contrasted with reported sequences in other species, the count of *Aux/IAA* family members surpasses that of maize (31) [26], *Prunus mume* (19) [27] and *Dendrobium ofcinale* (14) [28]. Conversely, the *ARF* family member count exceeds *Arabidopsis thaliana* (23) [29], bamboo (24) [6], and *Ginkgo biloba* (15) [17], yet remains lower than *Triticum aestivum* (67) [30], *Medicago truncatula* (40) [31], and soybean (51) [22]. Among them, the numbers of *Aux/IAA* and *ARF* genes in *P. simonii*, *Salix suchowensis*, and *P. trichocarpa* are the closest [21,32,33], indicating that the two gene families are relatively conserved within the Salicaceae family. Based on the phylogenetic trees constructed for *PsIAA* and *PsARF* families, all predicted *PsIAA* and *PsARF* family members can be classified into three major clusters (A, B, and C), which is consistent with the classification in *S. suchowensis*. 

According to the analysis of conserved domains, it was observed that PsIAA proteins exhibited relatively higher conservation, particularly in domain II and domain Ⅲ. However, four proteins (PsIAA20, PsIAA32, PsIAA8, and PsIAA30) lack domain I, which may lead to reduced recruitment of TPL proteins, leading to enhanced transcriptional activity of downstream *ARF* genes, consistent with previous research findings [3]. Furthermore, all PsARF proteins possessed a characteristic DNA-binding domain (DBD) and middle region (MR). The sequence and amino acid composition of the protein’s MR determine whether ARF acts as a transcriptional activator or a repressor [34]. In *Arabidopsis*, studies have revealed that *ARF1* and *ARF2*, which contain enriched proline (P), serine (S), and threonine (T) in their MRs, act as repressors, while *ARF5*, *ARF7*, and *ARF8*, which contain enriched glutamine (Q) in their MRs, function as activators [35]. Additionally, it was discovered that more than half of the PsARF proteins lack the C-terminal domain (CTD), suggesting that their activity may be regulated through interactions with other transcription factors rather than with Aux/IAA proteins [36]. This demonstrates their insensitivity to the plant hormone auxin. Therefore, we hypothesize that these PsARF proteins may be involved in plant development and growth in a manner independent of auxin interactions.

The exploration of gene structure and protein motif analysis within *PsIAAs* and *PsARFs* will facilitate an improved comprehension of their unique functions in development and growth processes [37]. The observation revealed that the majority of *PsIAAs* and *PsARFs* shared comparable exon–intron configurations within corresponding phylogenetic clusters. However, certain members displayed distinct structures, indicating variations within the same gene family. Of note, a compelling discovery emerged: the *PsARF* family displayed a notably elevated count and length of exons and introns in comparison to the *PsIAA* family. Gene characteristics stand as a primary factor influencing alternative splicing (AS), with intron retention emerging as the prevailing type of AS event in plants [6]. Research indicates that intron retention accounts for approximately 60%, 40% and 33% in *A. thaliana* [38], Zea mays [39], and *Oryza sativa* [40] of AS events, respectively. Moreover, it has been reported that 36% of genes in the poplar genome undergo alternative splicing, and the frequency of AS events increases with a higher number of exons [41]. Consequently, we postulate that the likelihood of AS events occurring in *PsARF* genes is greater than in *PsIAA* genes, and intron retention is anticipated to prevail as the dominant type of AS event in *P. simonii*. Additionally, the consistent distribution of conserved motifs among PsIAA and PsARF proteins within the same subfamily suggests that these members of the IAA and ARF protein families may share similar functions in plant growth, development, and stress response. The overall similarity in gene structures and motif compositions among *PsIAAs* and *PsARFs* in each group reinforces the grouping characteristics identified in the phylogenetic analysis. Through the chromosomal localization of genes, it was observed that chromosome 2 has the highest abundance, containing ten *PsIAA* and *PsARF* genes, while the largest chromosome, Chr1, only harbors seven *PsIAA* and *PsARF* genes. This indicates that the number of IAA and ARF family genes on each chromosome is not dependent on the size of the chromosome. Prior studies have demonstrated that gene duplication plays a pivotal role in driving adaptive evolution in plants. Numerous plants have experienced evolutionary occurrences such as fragmentations, tandem duplications, and whole-genome duplications, exerting a substantial influence on the establishment and rapid proliferation of gene families [42]. In the *PsIAA* family, a total of five pairs of tandemly duplicated gene pairs were identified, indicating that tandem duplication may have played a major role in the evolution of the *PsIAA* family. Furthermore, we did not identify any tandemly duplicated gene pairs in the *PsARF* family, indicating that the expansion of *PsARFs* is likely primarily driven by segmental duplications and whole-genome duplications, with tandem duplications not being involved in the amplification process of the *PsARF* family. Moreover, in the identified *PsIAA* and *PsARF* families, we obtained 36 and 30 syntenic gene pairs, respectively, through synteny analysis. This suggests that the interior of *P. simonii* may have undergone multiple selective evolutionary directions. However, inter-species collinearity analysis revealed that the *PsARF* family has a smaller number of syntenic gene pairs with other species, compared to the *PsIAA* family. This indicates that there may be differences in the evolution of different gene families in *P. simonii*. The occurrence of tandem duplications in the PsIAA family may have played a promoting role in this phenomenon.

Protein–protein interactions assume a pivotal role in plant developmental processes, notably in phenomena such as auxin response signaling, where interactions between ARF and Aux/IAA proteins mediate the cascade [9]. Therefore, studying the interactions between IAA and ARF proteins in *P. simonii* is of significant importance. In this study, the protein–protein interaction (PPI) network includes 564 interaction combinations between 28 *PsIAAs* and 30 *PsARFs*. In *Arabidopsis*, the integration of co-expression data with PPI data has identified 213 specific interactions between 19 *ARFs* and 29 *Aux/IAAs* [43]. Within bamboo, the PPI network has elucidated a total of 202 interaction combinations involving 12 *PeARFs* and 15 *PeIAAs* [6]. This indicates a complex regulatory network of interactions between *ARFs* and *Aux/IAAs*, providing the basis for the response to auxin and contributing to plant growth. Moreover, the interaction network revealed the presence of two pivotal hub genes, *PsIAA7* and *PsARF18*. These genes hold potential as promising candidates for investigating tissue development and formation within *P. simonii*.

In this study, we performed a comprehensive analysis of cis-regulatory elements in the promoters of each *PsIAA* and *PsARF* gene. Among these elements, many were found to be associated with plant growth, and development and abiotic stress responses. These findings not only support the molecular mechanisms attributed to *IAA* and *ARF* genes, as previously outlined in investigations of cellular signaling [21,44,45], but also offer insights that could aid in discerning the biological roles of *PsIAAs* and *PsARFs*. Analyzing the expression profiles of *Aux/IAA* and *ARF genes* in different tissues helps to understand the dynamics of gene expression during the growth and development processes in *P. simonii*. The results indicate that many *PsIAA* and *PsARF* genes exhibit higher expression levels in different tissues, suggesting their indispensable roles during various growth stages of *P. simonii*. Conversely, a minor subset of *Aux/IAA* and *ARF* members displayed distinct tissue-specific expression patterns, implying the potential existence of specialized functions for these genes. In contrast to other tissues, *PsIAA15*, *PsIAA21*, and *PsARF22* exhibited distinctly elevated expression levels exclusively in leaves, suggesting their possible involvement in leaf development processes. Prior research has documented upregulation of genes such as *BrIAA1*, *-6*, *-32*, and *-33* in leaves of *Brassica rapa* [46], and it was also found that genes such as *PeIAA3*, *-10*, *PeARF1*, and *-17* were predominantly expressed in bamboo leaves, promoting leaf development [6]. These findings are consistent with the results of our study. The expression of *PsIAA22* and *PsARF4* is significantly higher in the roots compared to other tissues. A comparable expression pattern was likewise identified in *Medicago truncatula* [31], signifying the involvement of these genes in root regulation. Fascinatingly, upon juxtaposing our findings with those documented by Ali, Shahid, et al. [22], we observed that a majority of *PsIAAs* and *PsARFs* (e.g., *PsIAA7*, *PsIAA29*, *PsARF2* and *PsARF26*) displayed relatively heightened expression levels across various tissues, which is a common observation in research. This may indicate that these genes are likely crucial for plant growth and development, playing essential roles across multiple tissues.

Although auxin has traditionally been acknowledged as a primary hormone responsible for promoting growth, an increasing body of evidence suggests that auxin also plays a role in stress responses by interacting with various other signaling pathways [47,48]. Furthermore, we found that many *PsIAAs* and *PsARFs* may be involved in responses to abiotic stress, as their gene expression levels were altered under stress conditions. Compared to other stresses, *PsIAA8*, *PsIAA26*, and *PsARF15* exhibited high levels of response to cold stress. *PsIAA12*, *PsIAA15*, *PsARF31*, and *PsARF14* showed extensive responses to cold and heat stresses. It is noteworthy that genes such as *PsIAA7*, *PsIAA18*, *PsIAA32*, *PsARF13*, and *PsARF23* responded to cold, heat, and salt stresses, with their gene expression levels being relatively high, indicating that they may be key regulatory genes in stress responses. Similarly, in other plants, it has been found that *IAA* and *ARF* genes are induced by stress treatments. All 20 *StARFs* showed strong upregulated expression under salt stress and mannitol treatment in the potato [48]. *SbIAA1*, *SbIAA26*, and *SbARF3* exhibited significant upregulation at the transcript level after salt stress in *Sorghum bicolor* [49]. Furthermore, in *P. trichocarpa*, 15 *PtrARF* genes were found to be potentially involved in responses to abiotic stress, and these findings are consistent with our research results. Noteworthily, *PsIAA7* exhibited higher expression levels in various tissues and under stress conditions, suggesting that it may act as a positive regulatory gene in the modulation of tissue development and stress resistance. *PsIAA7* holds the potential to be considered as a candidate gene for future studies on the development of *P. simonii*. Additionally, we conducted RT-qPCR analysis to validate the RNA-seq results, focusing on the expression profiles of the *PsIAA* and *PsARF* genes.

By conducting transient expression analysis of the *PsIAA7*–GFP fusion protein, we noted the localization of *PsIAA7* within the nucleus. Certain studies have demonstrated that specific members of the *Aux/IAA* gene family exhibit nuclear localization. For instance, in Solanaceae plants, it has been discovered that *SlIAA2*-*3*, *SlIAA8*, *SlIAA12*, *SlIAA15*, *SlIAA17*, and *SlIAA22-23* are localized in the cell nucleus [50]. Similarly, in *Dendrobium ofcinale*, *DoIAA1*, *DoIAA6*, *DoIAA10*, and *DoIAA13* have also been found to be located in the cell nucleus [28]. Our research findings are consistent with previous reports and confirm the involvement of *PsIAA7* in the regulation of auxin in the nucleus of *P. simonii*.

## 4. Materials and Methods

### 4.1. Identification of Aux/IAAs and ARFs in P. simonii

For the purpose of discovering *IAA* and *ABF* gene family members within the *P. simonii* genome, complete protein sequences of IAA and ABF from *A. thaliana* were acquired through the TAIR database (https://www.arabidopsis.org/ (accessed on 9 June 2023). Additionally, the high-quality chromosome-level *P. simonii* reference genome with a contig N50 of 24 Mb and a BUSCO complete gene percentage of 98.9% was prepared for subsequent analysis by integrating PacBio, Oxford Nanopore Technologies (ONT), and high-throughput chromosome conformation capture (Hi-C) data. First, PsIAA and PsARF protein sequences were identified using the BlastP program (e-value, 1 × e^−5^) of TBtools [51] with *A. thaliana* IAA and ABF proteins as query sequences. Duplicate sequences were removed from the search results and the assumed member sequences obtained. Subsequently, the conserved domains of candidate protein sequences were further identified using NCBI “batch Web CD-Search Tool” (https://www.ncbi.nlm.nih.gov/Structure/bwrpsb/bwrpsb.cgi (accessed on 20 June 2023) to detect each candidate protein as an Aux/IAA or ARF protein. Lastly, the online tools SMART (http://smart.embl-heidelberg.de/smart/set_mode.cgi (accessed on 4 July 2023) and Pfam (http://pfam-legacy.xfam.org/ (accessed on 18 July 2023) were used to conduct further verification of the presence of conserved domains in the search results, ultimately identifying the candidate *PsIAA* and *PsARF* genes. Utilizing TBtools software (version 2.008), the biochemical characteristics of each PsARF and PsIAA protein, encompassing amino acid count, molecular weight, isoelectric point (pI), and instability index, were subjected to analyses.

### 4.2. Phylogenetic Analyses of Aux/IAAs and ARFs in P. simonii

To explore the evolutionary relationship of *PsIAAs* or *PsARFs*, two phylogenetic trees were generated by utilizing all candidate protein sequences. The Muscle program with default parameters from MEGA11.0 software were used for multi-sequence alignment analysis [52]. Subsequently, phylogenetic trees were created using the neighbor-joining (NJ) approach with MEGA11.0 software (with 1000 replications for bootstrapping). To enhance visual presentation, the ultimate phylogenetic tree of the *PsIAA* and *PsARF* families was embellished and annotated using the online resource Interactive Tree of Life (iTOL) (https://itol.embl.de/ (accessed on 23 July 2023).

### 4.3. Chromosomal Localization and Synteny Analysis of PsIAA and PsARF Genes

The TBtools software (version 2.008) was utilized to pinpoint the starting and ending positions of *PsIAAs* and *PsARFs* on chromosomes in *P. simonii*. Based on this positional information, the online tool MG2C [53] was used to visualize the chromosomal location image of *PsIAAs* and *PsARFs*. Based on the physical location information of genes on chromosomes, the *PsIAA* and *PsARF* genes were re-named (*PsIAA1*-*PsIAA33*, and *PsARF1*-*PsARF35*). To further analyze the intra-species synteny of *PsIAA* and *PsARF* genes, the syntenic gene pairs of *PsIAAs* and *PsARFs* were identified through MCscanX analysis, and the synteny circos plot was visualized using “Advanced Circos” program from TBtools software (version 2.008). Moreover, the “One step MCScanX” program in TBtools software (version 2.008) was utilized for the syntenic analysis of *AUX/IAA* and *ARF* genes in *P. simmonii*, *P. trichocarpa*, and *A. thaliana*, and the results were visualized using the “Multiple Synteny Plot” program.

### 4.4. Gene Structure, Conserved Motif and Promoter Cis-Elements Analysis

The “Gene Structure View program” from TBtools was employed to visualize the exon/intron structure of the *PsIAA* and *PsARF* genes, using the genomic structure information (GFF) and gene ID data. The conserved motif configurations within the proteins encoded by PsIAAs and PsARFs were examined utilizing the online application Multiple Em for Motif Elicitation (MEME Version 5.5.3), accessible at http://memesuite.org/tools/meme (accessed on 28 July 2023). The parameter settings used were a maximum number of motifs set to 10, and a maximum width of 50. Furthermore, the promoter region sequences of *PsIAA* and *PsARF* genes, spanning 2000 bp upstream of the translational start site (ATG), were extracted from the *P. simonii* genome. Forecasting cis-elements within the promoters was achieved through the utilization of the PlantCARE tool (https://bioinformatics.psb.ugent.be/webtools/plantcare/html/ (accessed on 3 August 2023), while the visual representation of these cis-elements was generated using TBtools. The ggplot R package is used to construct corresponding heatmaps of promoter cis-elements [54].

### 4.5. Plant Materials and Sample Collections

The adult clones of *P. simmonii* used for this study were cultivated at the Forestry and Grassland Science Research Institute of Tongliao City, Inner Mongolia Autonomous Region, China (44°01′53″ N 121°59′33″ E). On a sunny day, samples of different tissues and developmental stages, including terminal buds (NT), axillary buds (NB), leaves (NL), one-year-old stems (NS), phloem (NP), and roots (NR) were collected. Upon collection, all samples were promptly submerged in liquid nitrogen and stored at a temperature of -80 °C for future analyses, particularly for transcriptome sequencing. To ensure the robustness and reliability of all samples, three repeat samples were collected for different tissues.

### 4.6. Expression Analysis and Interaction Network Construction of PsIAA and PsARF Genes

The RNA-seq profiles of various tissues of *P. simmonii* were used to acquire the expression values of *PsIAA* and *PsARF* genes, which were then visualized through heat maps generated using TBtools. Moreover, to analyze the expression patterns of *PsIAA* and *PsARF* genes under various stress conditions, we retrieved the raw RNA-seq reads from the NCBI sequence read archive (SRA) database (accession number: SRS1866875, SRR7686816, and SRS1890707), which included data from heat stress, cold stress, and salt stress treatments. After implementing quality control, alignment, and quantitative analysis procedures, we derived a count matrix. This matrix was subsequently employed to compute the expression levels of *PsIAA* and *PsARF* genes. Afterwards, the data was normalized and utilized to generate an expression heatmap for all *PsIAA* and *PsARF* genes associated with stress response. Furthermore, the function prediction of 33 PsIAA and 35 PsARF protein sequences was accomplished by constructing protein–protein interaction (PPI) networks using the STRING website (https://www.string-db.org/ (accessed on 13 August 2023). The obtained network files were visualized using Gephi software (version 0.9.2) [55].

### 4.7. RNA Extraction and qRT-PCR Analysis

The samples’ total RNA was isolated using the plant total RNA extraction kit from Takara (Beijing, China). To validate the outcomes of our earlier expression profiles, we chose a set of 12 pivotal DEGs and performed RT-qPCR analysis using designated primers. Utilizing TBtools software (version 2.008), we designed dedicated primers for the genes, with *Actin* serving as the internal reference gene [56] (Appendix A). For qRT-PCR analysis, the ABI 7500 RT PCR system was utilized. The 2^-∆∆Ct^ method was employed to calculate the relative expression levels of the selected gene [57]. Three biological replicates were used for gene expression analysis.

### 4.8. Analysis of Subcellular Localization of PsIAA Proteins

The Prime STAR HS (Premix), R040 (Takara Biotech, Beijing, China) was employed for PCR amplification to obtain the complete coding sequence (CDS) of *PsIAA7*, excluding the termination codon (Appendix A). For constructing the expression vector, the pCAMBIA1300 plasmid served as the foundational framework, harboring a 35S promoter. The vector was modified with the inclusion of the GFP green fluorescent protein as a reporter gene, leading to the formation of the pCAMBIA1300-35S:*PsIAA7*-GFP fusion expression construct. The primers utilized for gene cloning and vector assembly are detailed in Appendix A. Upon completion of the *PsIAA7*–GFP vector construction, it was introduced into Agrobacterium tumefaciens GV3101 via the freeze–thaw method. Subsequently, the acquired suspension was introduced into the leaves of 5–6-week-old tobacco plants (*Nicotiana tabacum*) using the injection technique. After a growth period of 36–48 h in the absence of light, the expression of *PsIAA7*–GFP was assessed utilizing confocal laser scanning microscopy (Nikon C2-ER, Tokyo, Japan).

## 5. Conclusions

In the current investigation, we have successfully identified and categorized a total of 33 *Aux/IAA* and 35 *ARF* gene families within *P. simonii*. Our analysis encompassed a comprehensive exploration of various aspects, such as phylogenetic connections, gene architecture, evolutionary associations, protein–protein interactions, expression profiles, and subcellular distribution. The outcomes of our experimental endeavors underscore the pivotal roles played by specific members within these two gene families in the development of plant tissues as well as in responses to various stress conditions. Specifically, we discovered that the hub gene *PsIAA7* exhibits significantly higher expression levels across various tissues and stress conditions, making it a potential candidate gene for future research on *P. simonii* development. Our investigation has provided valuable resources for further exploring the biological functions of *PsIAA* and *PsARF* genes in *P. simonii*, as well as unraveling the regulatory mechanisms of the auxin-related pathways.

## Figures and Tables

**Figure 1 plants-12-03566-f001:**
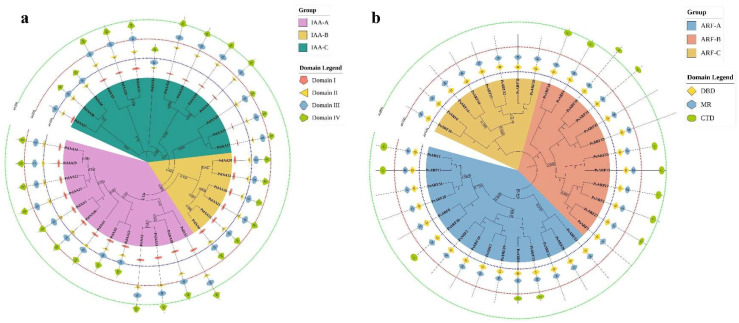
The phylogenetic relationships and conserved domain patterns in PsIAA and PsARF proteins. (**a**) The phylogenetic tree of PsIAA proteins is divided into three subfamilies, which are represented by green, yellow, and pink arcs. Domains I–IV are shaped with down pointing pentagram, left pointing triangle, horizontal hexagon, and left pointing pentagram from inside to outside. (**b**) The phylogenetic tree of PsARF proteins is also divided into three subfamilies, and the canonical domains DBD (DNA-binding domain), MR (middle region), and CTD (C-terminal dimerization domain) are represented by diamond, vertical hexagon, and ellipse, respectively. The lengths of PsIAA and PsARF proteins are visualized through a triad of concentric dashed circles, each adorned with distinctive hues.

**Figure 2 plants-12-03566-f002:**
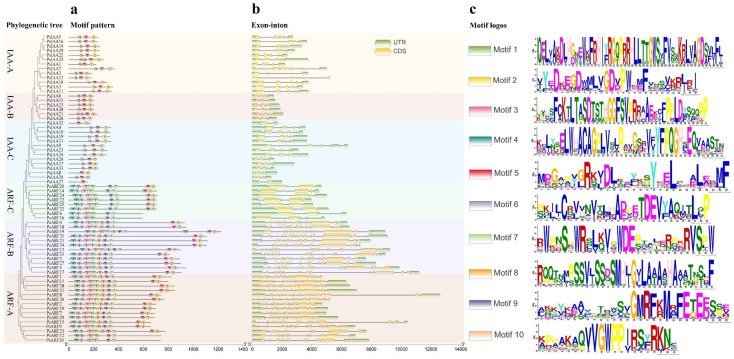
The gene structure and conserved motifs analysis of the *PsIAA* and the *PsARF* family. (**a**) The motif composition of PsIAA and PsARF proteins. Motifs 1–10 are displayed in different colored boxes. (**b**) The exon and intron structure of *PsIAAs* and *PsARFs*. The UTRs region, exons, and introns are depicted as green boxes, yellow boxes, and black lines, respectively. The ruler at the bottom indicates the length of the exon and intron segments. (**c**) The motif logos of PsIAA and PsARF proteins. On the left, the ten anticipated motifs are depicted as colored boxes, while on the right, the corresponding motif logos are displayed.

**Figure 3 plants-12-03566-f003:**
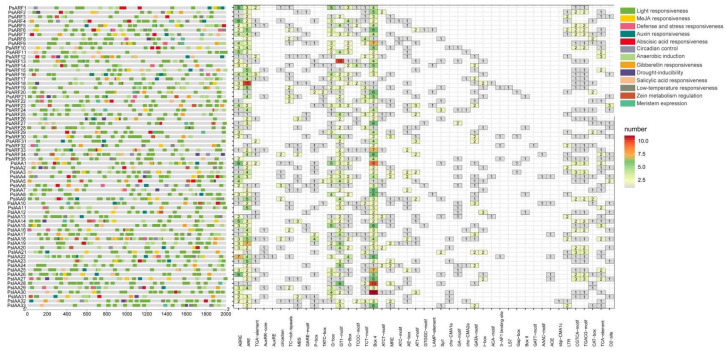
Analysis of cis-acting elements in *PsIAA* and *PsARF* promoters. The 2000 bp promoter region of each *PsIAA* and *PsARF* gene was analyzed for cis-elements. Distinctive hued boxes on the right correspond to cis-acting elements possessing diverse functions. The heat map shows the number of different cis-acting elements distributed in each gene.

**Figure 4 plants-12-03566-f004:**
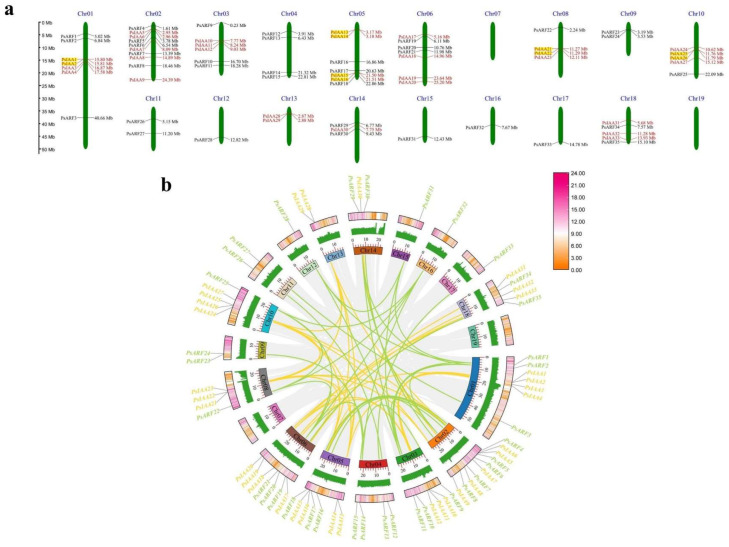
Chromosome distribution and synteny relationships of the *PsIAA* and *PsARF* families. (**a**) Chromosomal locations and duplicated gene pairs of distribution of *PsIAAs* and *PsARFs*. The genomic location of each gene is charted onto the chromosome according to its specific physical coordinates. The top indicates the number of chromosomes (Chr01–Chr19), with all gene names on the left and their corresponding physical distances on the right. The yellow shading represents tandemly duplicated gene pairs. (**b**) Circso plot of gene duplication of *PsIAA* and *PsARF* genes. The heatmap illustrates gene density distribution, while the histograms depict the GC ratio distribution along the chromosomes. Gray lines represent the syntenic blocks within the *P. simonii* genome, while the collinear pairs of *PsIAA* and *PsARF* genes are denoted by yellow and green lines, respectively.

**Figure 5 plants-12-03566-f005:**
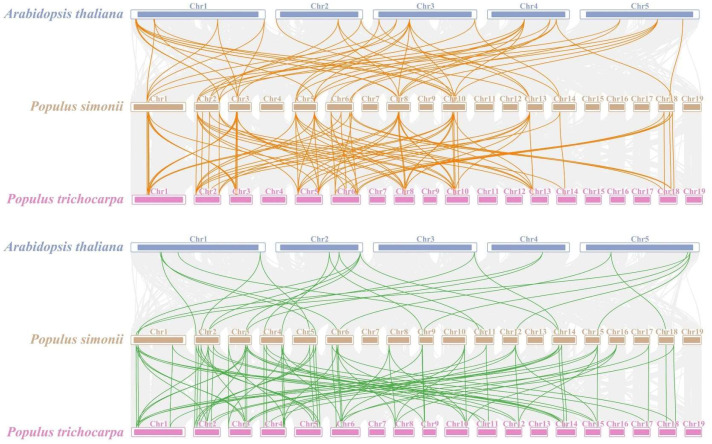
Synteny relationships of the *IAA* and *ARF* genes between *P. simonii*, *A. thaliana* and *P. trichocarpa*. The orange lines emphasize the syntenic pairs of *IAA* genes. The green lines represent homologous pairs of *ARF* genes, while the gray lines depict collinear segments in *P. simonii* that are orthologous to two plant genomes.

**Figure 6 plants-12-03566-f006:**
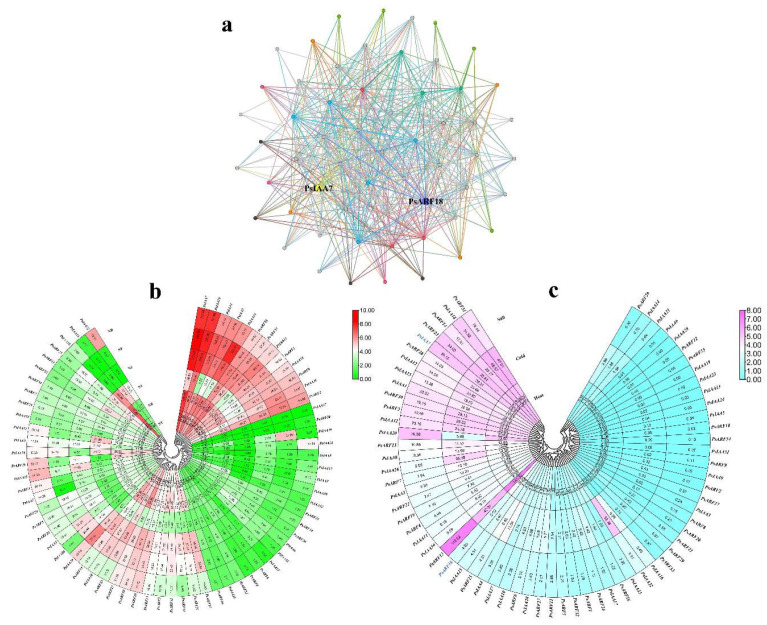
The interaction networks and expression patterns of *PsIAA* and *PsARF* genes. (**a**) The network contains 58 nodes (28 PsIAA protein and 30 PsARF protein) and 564 edges (interaction combinations). Network nodes symbolize proteins, and edges indicate interactions between proteins in the network. (**b**) Expression patterns of *PsIAA* and *PsARF* genes across different tissues. The heatmap illustrates the hierarchical clustering of *PsIAA* and *PsARF* genes in various tissues. The color spectrum transitioning from green to red denotes varying expression levels, ascending from low to high. NT, terminal buds; NB, axillary buds; NL, leaves; NS, stems; NP, phloem; and NR, roots. (**c**) Expression patterns of *PsIAA* and *PsARF* genes under stress. The color gradient ranging from blue to pink signifies a spectrum of expression levels, progressing from low to high. Salt, salt stress; Cold, cold stress; and Heat, heat stress.

**Figure 7 plants-12-03566-f007:**
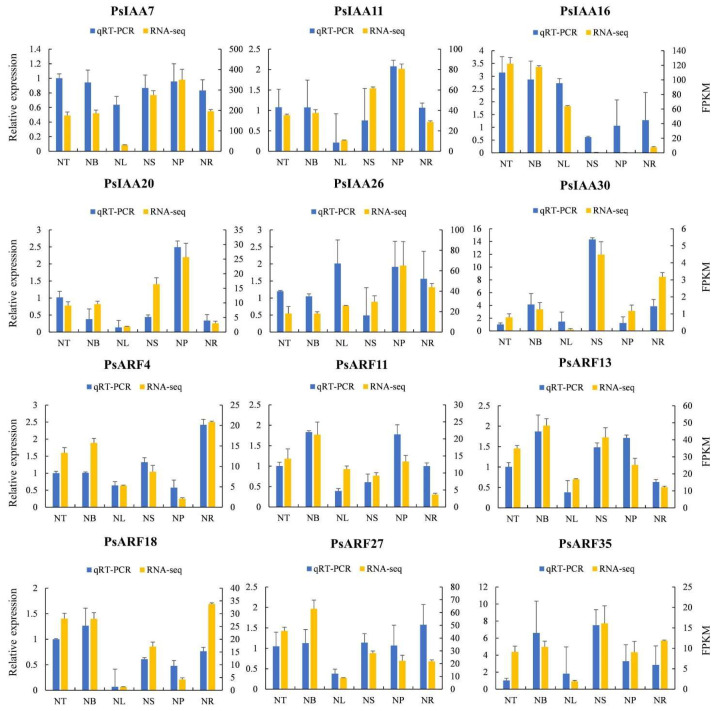
Gene expression patterns of RNA-seq and qPCR. The X-axis represents six different tissues. The Y-axis on the left represents the expression data of qRT-PCR. The Y-axis on the right represents the relative expression levels of genes validated by RNA-seq. The error bars represent standard error.

**Figure 8 plants-12-03566-f008:**
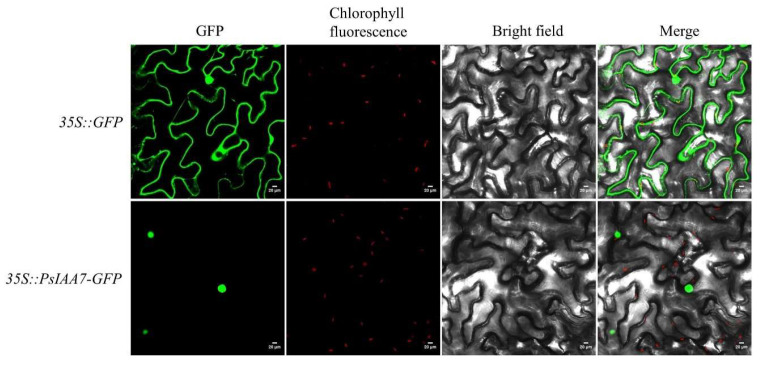
Subcellular localization analysis of the PsIAA7 protein. Subcellular localization of PsIAA7 was examined in tobacco leaves expressing green fluorescent protein (GFP). *35S::GFP*, positive control; *35S::PsIAA7-GFP*, PsIAA7 and GFP fusion protein; GFP, green fluorescence signal; Chlorophyll fluorescence, chlorophyll fluorescence signal; Bright field, white light signal; Merge, superposition of different fluorescence signals. Bars = 20 μm.

## Data Availability

All data generated or analyzed during this study are included in this published article.

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
