# Peer review of "Unraveling the Guardians of Growth: A Comprehensive Analysis of the Aux/IAA and ARF Gene Families in Populus simonii"

_plants, 2023, doi:10.3390/plants12203566_

Round 1

Reviewer 1 Report

I checked your manuscript and described comments below.

Populus simonii is an important street tree around the world.

This paper does a good job of analyzing Aux/IAA and ARF Gene Families of Populus simonii.

I think it would be better to correct the following points.

1.       It would be better to include the protein amino acid sequence information in Table S1. With the current information, other researchers cannot confirm this.

2.       It would be better to use the DNA sequence of the gene as a supplementary file.

3.       There is no bootstap value in the phylogenetic results. It should be described in the diagram.

4.       The references for MEGA11 used in phylogenetic analyzes are below.

MEGA11: Molecular Evolutionary Genetics Analysis version 11

Tamura K, Stecher G, and Kumar S (2021)

Molecular Biology and Evolution (https://doi.org/10.1093/molbev/msab120):

I don't think this paper has any major mistakes or grammatical problems.

Author Response

Reviewer 1:

I checked your manuscript and described comments below. Populus simonii is an important street tree around the world. This paper does a good job of analyzing Aux/IAA and ARF Gene Families of Populus simonii. I think it would be better to correct the following points.

  1. It would be better to include the protein amino acid sequence information in Table S1. With the current information, other researchers cannot confirm this.

Response:

Thanks for your comments. It is really true as your suggested. We are very sorry for our negligence. We have made correction according to your comments.

Please see Table S1 in the tracked manuscript.

Special thanks to you for your good comments.

  1. It would be better to use the DNA sequence of the gene as a supplementary file.

Response:

Thanks for your comments. It is really true as your suggested. We are very sorry for our negligence. We have made correction according to your comments. We have placed the DNA sequence of PsIAA7 in a supplementary file.

Special thanks to you for your good comments.

  1. There is no bootstap value in the phylogenetic results. It should be described in the diagram.

Response:

Thanks for your comments. It is really true as your suggested. We are very sorry for our negligence. We have made correction according to your comments.

Please see Figure 1 in the tracked manuscript.

Special thanks to you for your good comments.

  1. The references for MEGA11 used in phylogenetic analyzes are below.

MEGA11: Molecular Evolutionary Genetics Analysis version 11 Tamura K, Stecher G, and Kumar S (2021) Molecular Biology and Evolution (https://doi.org/10.1093/molbev/msab120):

Response:

Thanks for your comments. It is really true as your suggested. We are very sorry for our negligence. We have made correction according to your comments.

Special thanks to you for your good comments.

I don't think this paper has any major mistakes or grammatical problems.

Response:

Thank you for your comments. These comments are all valuable and very helpful for revising and improving our paper, as well as the important guiding significance to our researches.

Reviewer 2 Report

Dear authors, your study sheds light on two gene families Aux/IAA and ARF in Populus simonii, thereby contributing to genomics studies in tree species. Please, check the description in Figure S1.

Author Response

Reviewer 2:

Dear authors, your study sheds light on two gene families Aux/IAA and ARF in Populus simonii, thereby contributing to genomics studies in tree species. Please, check the description in Figure S1.

Response:

Thanks for your comments. It is really true as your suggested. We are very sorry for our negligence. We have made correction according to your comments.

Please see Figure S1 in the tracked manuscript.

Special thanks to you for your good comments.

Reviewer 3 Report

Thank you for your invitation to review the review article titled “Unraveling the Guardians of Growth: A Comprehensive Analysis of the Aux/IAA and ARF Gene Families in Populus simonii The article is interesting since it discuss scientific literature indicating some limitations of phytohormones methods. The article is clear in the display of backgrounds as well as in results. Moreover, it also shows a relevant literature research even if the only 11 on 66 cited articles have been published in the last 5 years. But there is some minor revision :

1-    A revision of language is also recommended. The article is suitable for publication

2-    Hypothesis of study should be elaborated in more detail

3-    Introduction must be improved by more information

4-    Results section is clear section

5-    Materials and methods is very good written

6-    Discussion must be improved by more information

7-    References  must be updated by new references about 2022 and 2023

Moderate editing of English language required

Author Response

Reviewer 3:

Thank you for your invitation to review the review article titled “Unraveling the Guardians of Growth: A Comprehensive Analysis of the Aux/IAA and ARF Gene Families in Populus simonii” The article is interesting since it discuss scientific literature indicating some limitations of phytohormones methods. The article is clear in the display of backgrounds as well as in results. Moreover, it also shows a relevant literature research even if the only 11 on 66 cited articles have been published in the last 5 years. But there is some minor revision :

1- A revision of language is also recommended. The article is suitable for publication

Response:

We appreciate your constructive feedback on the language quality of our paper. Recognizing the importance of clear and precise communication, we took your suggestion seriously and found native English speakers to revise and enhance the manuscript.

Special thanks to you for your good comments.

2- Hypothesis of study should be elaborated in more detail

Response:

Thanks for your comments. It is really true as your suggested. We have made correction according to your comments.

Special thanks to you for your good comments.

3- Introduction must be improved by more information

Response:

Thanks for your comments. It is really true as your suggested. We have made correction according to your comments.

Please see the introduction in the tracked manuscript.

Special thanks to you for your good comments.

4- Results section is clear section

Response:

Special thanks to you for your good comments.

5- Materials and methods is very good written

Response:

Special thanks to you for your good comments.

6- Discussion must be improved by more information

Response:

Thanks for your comments. It is really true as your suggested. We have made correction according to your comments.

Please see the Discussion in the tracked manuscript.

Special thanks to you for your good comments.

7- References must be updated by new references about 2022 and 2023

Response:

Thanks for your comments. It is really true as your suggested. We are very sorry for our negligence. We have updated the references according to your comments.

Special thanks to you for your good comments.